# No Identity, no problem: Motion through detection for people tracking

**Martin Engilberge**  *martin.engilberge@epfl.ch*
*Computer Vision Laboratory, EPFL*

**F. Wilke Grosche**  *friedrich.grosche@epfl.ch*
*Computer Vision Laboratory, EPFL*

**Pascal Fua**  *pascal.fua@epfl.ch*
*Computer Vision Laboratory, EPFL*

**Reviewed on OpenReview:** *https://openreview.net/forum?id=ogEM2H9IGK*

## Abstract

Tracking-by-detection has become the *de facto* standard approach to people tracking. To increase robustness, some approaches incorporate re-identification using appearance models and regressing motion offset, which requires costly identity annotations. In this paper, we propose exploiting motion clues while providing supervision only for the detections, which is much easier to do.

Our algorithm predicts detection heatmaps at two different times, along with a 2D motion estimate between the two images. It then warps one heatmap using the motion estimate and enforces consistency with the other one. This provides the required supervisory signal on the motion without the need for any motion annotations. In this manner, we couple the information obtained from different images during training and increase accuracy, especially in crowded scenes and when using low frame-rate sequences.

We show that our approach delivers state-of-the-art results for single- and multi-view multi-target tracking on the MOT17 and WILDTRACK datasets.

## 1 Introduction

Multi-target tracking has been extensively studied (Ciaparrone et al., 2020; Yilmaz et al., 2006; Smeulders et al., 2014), and research has now pivoted towards challenges such as handling crowded scenes, low frame rate sequences, and fast-moving objects.

Most current approaches rely on the tracking-by-detection paradigm (Andriluka et al., 2008), which requires strong object detectors. Linking the detections into trajectories is entrusted to algorithms that do not take the images into consideration anymore, such as Pirsiavash et al. (2011); Berclaz et al. (2011); Wang et al. (2019). Unfortunately, performance suffers in crowded scenes, especially when the video frame rate is low (Bergmann et al., 2019). Fig. 1 illustrates this. Motion prediction can be used to alleviate this problem. For example, the popular tracking-by-regression approach of Zhou et al. (2020); Xu et al. (2020) does this by predicting people's motion as a 2D motion offset.

However, motion supervision is often derived from identity annotations, which can be costly. In this paper, we seek to exploit the motion information *without* having to provide identity or motion annotations, relying solely on detection annotations that are much easier to obtain. To this end, we introduce a differentiable approach to constructing a future detection heatmap by displacing a current heatmap using a 2D offset map that represents motion. Given a pair of images acquired at times $t$ and $t+1$, we predict detection heatmaps at both times, along with an offset map. We use the offset map to warp the heatmap estimated at time $t$ to

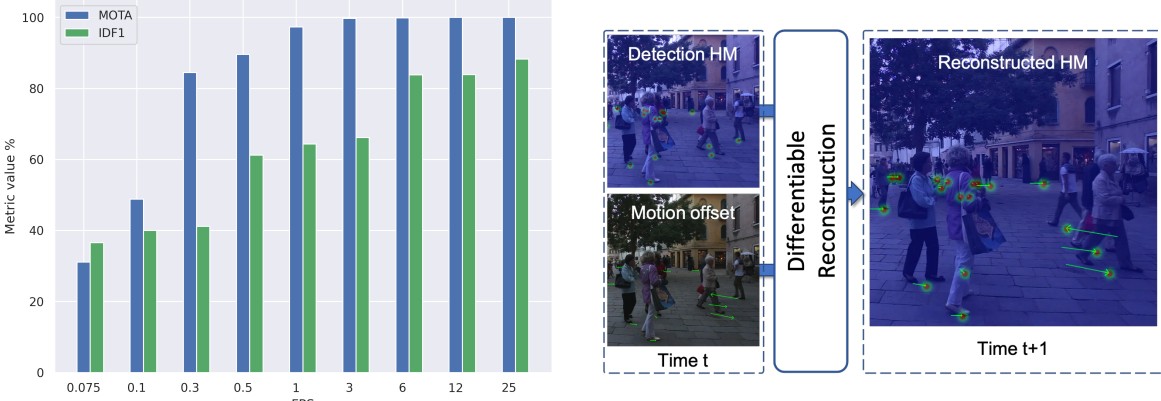

Figure 1: **Predicting human motion. Left:** We use muSSP Wang et al. (2019) to link detections at different frame rates. We plot the MOTA and IDF1 metrics as a function of the frame rate. Below 3FPS, the degradation becomes severe. **Right:** Our model estimates a detection heatmap at time $t$ and predicts the motion of objects between $t$ and $t+1$. The offsets are used to warp the heatmap into a prediction at time $t+1$ and we enforce consistency between that prediction and the one estimated by the network using the image acquired at time $t+1$.

enforce consistency between the result and the one estimated at time $t+1$. In this manner, we couple the information obtained from different images, which increases robustness.

The proposed motion estimation method can be used in conjunction with a wide variety of detectors and trackers. The supervision for detection can originate from ground-truth detections when trained jointly with a detector, or from the output of a pre-trained detector, if trained separately. Crucially, in either case, motion offsets are supervised without *any* additional annotations.

We use the challenging single-view MOT17 dataset (Milan et al., 2016) and multi-view WILDTRACK dataset (Chavdarova et al., 2018) to demonstrate our model's ability to predict accurate human motion without requiring any motion annotations. On MOT17 we outperform the state-of-the-art tracking method Bytetrack (Zhang et al., 2022) by a significant margin in low frame-rate scenario. On WILDTRACK we surpass recent multi-view detection and tracking techniques (Engilberge et al., 2023; Teepe et al., 2024; Cheng et al., 2023). Code can be found at `https://github.com/cvlab-epfl/noid-nopb`.

## 2 Related Work

Tracking is a core task of computer vision and has applications in a wide range of domains such as surveillance, autonomous driving, the medical field or the autonomous sales space. Despite having been widely studied (Ciaparrone et al., 2020; Yilmaz et al., 2006; Smeulders et al., 2014), multi-object tracking can still be challenging. In particular, self-occlusions are frequent in crowded environments and can result in missed detections and identity switches. Furthermore, due to hardware constraints, some applications can only operate at low frame rates, making association across frames difficult. In this section, we briefly review existing approaches and discuss multi-view tracking, which we use to validate our approach.

### 2.1 Multiple Object Tracking (MOT).

Most current MOT approaches operate on videos and follow the tracking-by-detection (Andriluka et al., 2008) paradigm. This requires strong object detectors, such as Faster R-CNN (Ren et al., 2015; Girshick, 2015), DPN (Felzenszwalb et al., 2010) or more recently anchor free models such as CenterNet (Zhou et al., 2019).

Given the detections in all frames of a video sequence, they must be grouped into trajectories that span the whole video sequence. Many approaches to doing this have been proposed. They can either be online (Breitenstein et al., 2009; Bewley et al., 2016; Wojke et al., 2017), where association is done frame-by-frame, or offline by generating tracks given detections throughout the entire video sequence (Berclaz et al., 2006; 2011; Wang et al., 2019). It has been shown (Bergmann et al., 2019) that simple association methods combined with strong detectors are able to handle most tracking scenarios and are on par with more complex methods. Similarly, Zhou et al. (2020) propose to use a greedy association mechanism combined with an anchor free detector regressing 2D motion offsets. In Voigtlaender et al. (2019), multiple object tracking and segmentation are combined by propagating segmentation masks using optical flow (Dosovitskiy et al., 2015; Teed & Deng, 2020).

Many of these association algorithms rely on a graph formulation. The graph nodes correspond either to all the spatial locations where an object can be present (Fleuret et al., 2008; Berclaz et al., 2011; BenShitrit et al., 2014) or only to detections (Jiang et al., 2007; Tang et al., 2015; Shu et al., 2012; Benfold & Reid, 2011; Cheng et al., 2023). The Successive Shortest Paths (SSP) (Pirsiavash et al., 2011) is a good representative of this class of techniques. It links detections using sequential dynamic programming. It was later extended with bounded memory and computation, which enables tracking in even longer sequences (Lenz et al., 2015). It was further optimized by exploiting the structures of the graphs formulated in multiple objects tracking problems (Wang et al., 2019). To deal with heavy occlusions, appearance models can be integrated into it.

To make further use of appearance, one can learn a deep association metric on a large-scale person re-identification dataset which enables reliable people tracking in long video sequences (Wojke et al., 2017). A discriminative appearance feature can also be learned using a hard identity mining triplet loss (Ristani & Tomasi, 2018), but it requires expensive identity annotations. Motion estimation has also been used in conjunction with deep learning (Liu et al., 2020; 2022; Engilberge et al., 2023) when modeling people densities and their evolution over time, but could only model limited motion ranges.

### 2.2 Multi-View Tracking.

Using multiple viewpoints is an effective way to increase robustness in crowded scenes featuring heavy occlusions. Some methods first extract monocular detections before projecting and matching them into a common frame (Xu et al., 2016; Fleuret et al., 2008) while others combine view aggregation and prediction (Baqué et al., 2017). Many recent ones (Hou et al., 2020; Song et al., 2021; Hou & Zheng, 2021; Engilberge et al., 2023;; Teepe et al., 2024) perform view alignment directly in the feature space. They assume the ground to be flat and project features onto the common ground plane. There, they are spatially aligned and can be easily aggregated to produce detection directly on the ground plane. The approach of (Engilberge et al., 2023) leverages properties of the ground plane by predicting weakly supervised human structure motion for tracking. We also leverage the benefit provided by such ground plane representations, but depart from previous work by predicting motion as an unconstrained 2D offset, allowing us to model and predict motion of arbitrary length.

## 3 Approach

Most recent single- and multi-view tracking approaches rely on simple association methods to link the output of strong single-frame object detectors. Typical constraints imposed on the linking process are limits on displacement length across frames (Engilberge et al., 2023; Bergmann et al., 2019), which are weak when the frame rate is high and inapplicable when it is low. To impose motion constraints while handling low frame rates and the potentially large frame-to-frame displacements they entail, we want to avoid such strict limits.

To this end, we introduce the approach depicted by Fig. 2. For all sets of images taken at consecutive time steps $t$ and $t+1$, we predict probabilities of detections at both time steps and a 2D displacement map from time $t$ to $t+1$. During training, we provide supervision for the detections and enforce consistency between the detections and displacements, which provides an additional supervisory signal without any additional annotations and increases performance, as will be shown Section 4.2

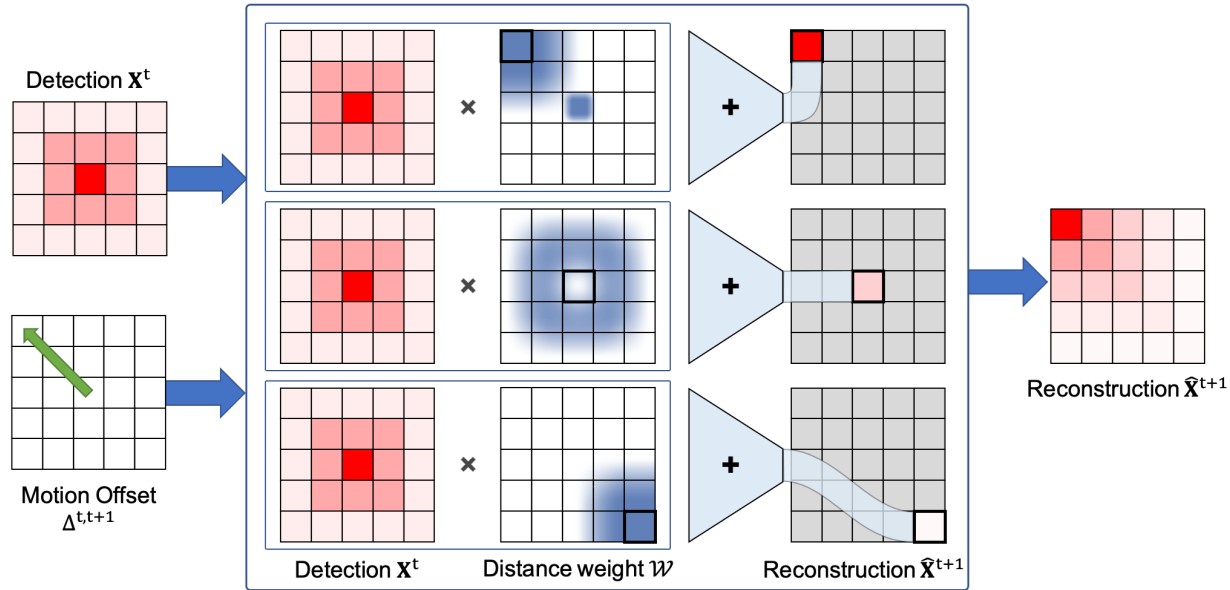

Figure 2: **Details of the proposed differentiable reconstruction from motion** Given a detection map at time $t$ and an offset map capturing motion of objects between time $t$ and time $t+1$ we reconstruct the detection map at time $t+1$. Each reconstructed pixel is a weighted sum of the detection of the previous time step, the weights are derived from the distance between the reconstructed location and the expected position of the previous locations after being moved by the offset. The example above illustrates the reconstruction for three locations, for the bottom one it is mainly unaffected by the offset and the distance weight is therefore a disc decreasing as the distance to the reconstructed location increases. For the middle reconstruction, the offset shows that the object has moved away from that location, therefore the corresponding weight for that location will be small. For the top reconstruction, the offset arrives at that location, therefore the contribution from the starting point of the offset will be high.

## 3.1 Formalism

We denote $\mathbf{I}_c^t$ an image drawn from camera $c \in [1, C]$ at acquisition time $t \in [1, T]$. Here $C \geq 1$ is the number of cameras and $T \geq 1$ is length of the video sequence. Let $\mathbf{P}_c^t$ be pairs of consecutive images taken at times $t$ and $t+1$ by camera $c$. We define as $\mathbf{P}^t$, the set of all pairs of frames taken at time $t$ and $t+1$.

For each camera $c$, let $\mathbf{K}_c$ be the intrinsic camera matrix and let $\mathbf{R}_c$ and $\mathbf{t}_c$ be the camera rotation and translation parameters which can be obtained by calibrating the cameras (Tsai, 1987). As in many other works (Hou et al., 2020; Engilberge et al., 2023), we will reason in a *ground plane* that is common to all views. We assume the ground to be flat and set its z-coordinate to 0. We can now define a homography $\mathbf{H}_c = \mathbf{K}_c[\mathbf{R}_c|\mathbf{t}_c]$ that relates points in the image plane $c$ to corresponding points in the ground plane. The assumption of a flat ground plane is valid in most crowd tracking scenarios but the approach can be generalised to complex topographies. The generalisation is performed by replacing the homography with a non-linear mapping that can be precomputed. We denote the ground plane as a 2D grid of size $w \times h$. We define $G = \{(x, y) \mid x \in [0 \ldots w] \text{ and } y \in [0 \ldots h]\}$ the set of all locations in the ground plane.

For every pair of frames $\mathbf{P}^t$, our network predicts a heatmap $\mathbf{X}^t \in [0, 1]^{w \times h}$ of the same dimension as the ground plane. This heatmap represents the probability that someone is present at a given ground plane location at time $t$. We also predict a corresponding displacement map $\Delta^{t,t+1} \in \mathbb{R}^{2 \times w \times h}$ that indicates where a person located at a given place at time $t$ is likely to be at time $t+1$. Each value of $\Delta^{t,t+1}$ is a 2D vector that represents the $x$ and $y$ components of the displacement of the person at that location. $\Delta^{t,t+1}$ can be used to compute a heat map $\hat{\mathbf{X}}^{t+1}$ as discussed in Section 3.2. These estimated values can then be compared to ground truth binary maps $\mathbf{X}_{\text{gt}}^t$ and $\mathbf{X}_{\text{gt}}^{t+1}$ that denote the actual presence of someone at any given location.

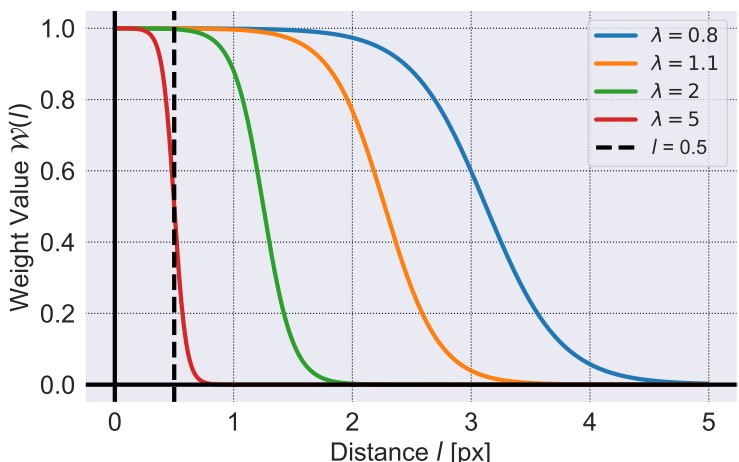

Figure 3: **Reconstruction weight function** During the reconstruction from detection and offset, the contributions of previous detections are reweighted using the function plotted above. With a value of $\lambda_r = 0.8$ a location 2 pixels away from the reconstructed location (distance accounting for motion offset) has a weight of 1 and is fully added to the reconstruction at that location. By varying $\lambda_r$ we control the trade-off between reconstruction accuracy and differentiability. Note that when $\lambda_r = 5$ the weight value is only one when the pixel distance is smaller than 0.5 (represented by the dashed, black line), during reconstruction this means that the detection in the previous time step only contributes to a single location in the next one.

## 3.2 Motion-Based Supervision

Given a frame $\mathbf{P}_t$, most existing methods compute $\mathbf{X}^t$ and $\mathbf{X}^{t+1}$ independently and are trained to make them as similar as possible to $\mathbf{X}^t_{\mathrm{gt}}$ and $\mathbf{X}^{t+1}_{\mathrm{gt}}$. By contrast, our approach estimates $\mathbf{X}^t$ and $\Delta^{t,t+1}$ and uses them to infer $\hat{\mathbf{X}}^{t+1}$. In effect, we force the estimated motion to be consistent with the observed probabilities of presence, which provides a supervisory signal on the predicted motion without having to provide motion annotations. Note that when we repeat the process on $\mathbf{P}_{t+1}$, we also force the $\mathbf{X}^{t+1}$ directly predicted by the network to be similar to $\mathbf{X}^{t+1}_{\mathrm{gt}}$. Thus, we also use the annotations as effectively as the earlier methods.

To implement this, the inference of $\hat{\mathbf{X}}^{t+1}$ from $\mathbf{X}^t$ and $\Delta^{t,t+1}$ must be differentiable. To this end, we implemented the following algorithm that derives a value at any given location by spatially global interpolation. Let $x^t_j$ and $\hat{x}^{t+1}_j$ be probabilities of presence at location $j$ of $\mathbf{X}^t$ and $\hat{\mathbf{X}}^{t+1}$.

We write

$$\hat{x}^{t+1}_j = \sum_{i \in G} x^t_i \times \mathcal{W}\Big(d\Big(j, i + \delta^{t,t+1}_i\Big)\Big), \tag{1}$$

$$\mathcal{W}(l) = \frac{1}{1 + e^{4\lambda_r l - 10}}, \tag{2}$$

where $\delta^{t,t+1}_i$ is the predicted displacement at location $i$ of $\Delta^{t,t+1}$ and $d$ is the Euclidean distance between coordinates $i$ and $j$. Note that $i$ and $j$ represent locations in the ground plane and consist of pairs $(x, y)$ of pixel coordinates.

$\mathcal{W}(l)$ is a weight function that decays with distance $l$ and $\lambda_r$ is a hyper-parameter that controls the speed of the decay. In Fig. 3 we plot this function for different $\lambda_r$. During training, lower values of $\lambda_r$ make convergence easier by allowing contribution from locations that are further apart. However, increasing $\lambda_r$ makes the reconstruction more accurate. For this reason $\lambda_r$ is initialized with a small value during training, which is then increased over time as the precision of the predicted offset improves.

We can now define a loss function

$$L_{\text{mot}} = \sum_t l_{\text{mot}}\left(\mathbf{P}^t\right) \tag{3}$$

$$l_{\text{mot}}\left(\mathbf{P}^t\right) = \|\hat{\mathbf{X}}^{t+1} - \mathbf{X}_{\text{gt}}^{t+1}\|^2 \, ,$$

whose minimization will enforce consistency between estimates of the probability of presence and the predicted motion.

### 3.3 Training Loss Function

When training our model, we minimize the composite loss function

$$L = L_{\text{mot}} + L_{\text{det}} + \lambda_{\text{fb}} L_{\text{fb}} + \lambda_{\text{se}} L_{\text{se}} \, , \tag{4}$$

where $L_{\text{mot}}$ is the motion-aware loss of Eq. 3 while $L_{\text{det}}$ is a detection loss and $L_{\text{fb}}$ and $L_{\text{se}}$ are regularization losses weighted by the scalars $\lambda_{\text{fb}}$ and $\lambda_{\text{se}}$, which we describe below.

#### 3.3.1 $L_{\text{det}}$ : Detection loss.

Detection heatmaps at time $t$ and $t+1$ are supervised using ground-truth detections. We write

$$L_{\text{det}} = \sum_t l_{\text{det}}\left(\mathbf{P}^t\right) \, , \tag{5}$$

$$l_{\text{det}}\left(\mathbf{P}^t\right) = \|\mathbf{X}^t - \mathbf{X}_{\text{gt}}^t\|^2 + \|\mathbf{X}^{t+1} - \mathbf{X}_{\text{gt}}^{t+1}\|^2 \, .$$

#### 3.3.2 $L_{\text{fb}}$ : Forward/Backward Loss.

For any $t$, we can reverse the order of the images in the frame $\mathbf{P}_t$ and, instead of estimating $\Delta^{t,t+1}$, estimate $\Delta^{t+1,t}$. In other words, we can reverse the motion and, if the network is appropriately trained, we should obtain a motion that is approximately the opposite of the original one. To enforce this, we define

$$L_{\text{fb}} = \sum_t l_{\text{fb}}\left(\mathbf{P}^t\right) \, , \tag{6}$$

$$l_{\text{fb}}(\mathbf{P}^t) = \sum_{i \in G} \left(\delta_i^{t,t+1} + \delta_j^{t+1,t}\right)^2 ,$$

where $j = i + \delta_i^{t,t+1}$.

#### 3.3.3 $L_{\text{se}}$ : Motion Spatial Extent Loss.

In practice, the ground truth maps are obtained by pasting Gaussians kernels where people are and zeros elsewhere. As a result, individual detections appear as Gaussian peaks. Since we train our networks to approximate these ground-truth maps, the same can be said about $\mathbf{X}^t$. To reliably move such peaks from their location at time $t$ to that at time $t+1$, the predicted motions should have roughly the same spatial extent. To encourage this, we define

$$L_{\text{se}} = \sum_t l_{\text{se}}\left(\mathbf{P}^t\right) \tag{7}$$

$$l_{\text{se}}(\mathbf{P}^t) = \frac{1}{N} \sum_p STD\left(\{\delta_i^{t,t+1} : i \in \mathcal{N}_p\}\right) \, ,$$

where $STD$ is the function computing the standard deviation, and $\mathcal{N}_p$ the neighborhood of ground truth point $p$ with the same radius as the ground truth Gaussian peaks.

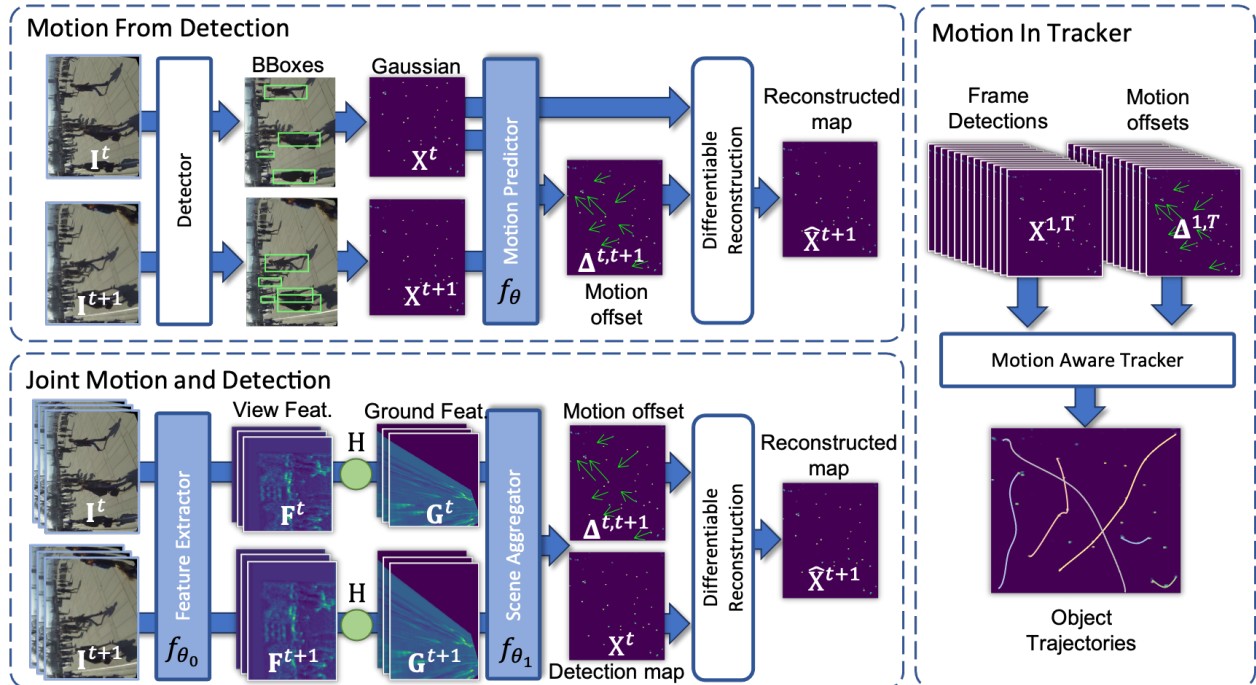

Figure 4: **Network Architectures.** Our approach to motion prediction is flexible and usable in conjunction with various detectors/trackers. Trainable components have a blue background, while static modules have a white background. **Top Left:** Single-view setup where the detector's serves as the primary training signal for the motion predictor. **Bottom Left:** Multi-view with simultaneous training for detection and motion. ResNet feature extractor produces feature maps $F$. These are projected onto a common ground via view homographies $\mathbf{H}_c$ resulting in $G$. These are combined, and passed to a scene aggregator producing an initial detection heatmap $\mathbf{X}^t$ and a motion offset map $\Delta^{t,t+1}$, which a differentiable reconstruction module uses to compute the next frame's detection map $\hat{\mathbf{X}}^{t+1}$. **Right:** The predicted motion can be leveraged by trackers to predict higher quality trajectories.

## 3.4 Model Architecture and Training

We use two different network architectures depicted by Fig. 4, one for the single-view case and the other for the multi-view one.

### 3.4.1 Single-View Model.

We extend the state-of-the-art YOLOX architecture (Ge et al., 2021) to process a pair of frames, $\mathbf{P}^t$, and to output an additional 2D displacement map, $\Delta^{t,t+1}$. Since YOLOX returns bounding boxes, we convert them to Gaussian heatmaps $\mathbf{X}^t$ and $\mathbf{X}^{t+1}$. They are then concatenated and fed through a ResNet (He et al., 2016) parametrized by weight $\theta$, which outputs a 2D displacement map $\Delta^{t,t+1}$. Finally, the displacement map and the heatmap $\mathbf{X}^t$ are fed to the prediction module of Section 3.2 to estimate $\hat{\mathbf{X}}^{t+1}$.

### 3.4.2 Multi-View Model.

The network we use is similar in design to the ones of Hou et al. (2020); Engilberge et al. (2023). As shown in Fig. 4, it takes as input the images of $\mathbf{P}^t$ and uses a ResNet He et al. (2016) parametrized by weight $\theta_0$ to extract features from all images from all cameras at both time steps independently. The feature maps are then projected onto a common ground plane using their respective image to ground plane homography $\mathbf{H}_c$, as defined in Section 3.1. The results are concatenated and processed by a scene aggregator parametrized by weight $\theta_1$ that outputs a detection probability map $\mathbf{X}^t$ and a 2D displacement map $\Delta^{t,t+1}$. Both are fed to the prediction module of Section 3.2 to estimate $\hat{\mathbf{X}}^{t+1}$.

### 3.4.3 Training

For both model $\mathbf{X}^t$, $\hat{\mathbf{X}}^{t+1}$, and $\Delta^{t,t+1}$ are used to compute the losses $L_{mot}$ of Eq. 3 and $L_{se}$ of Eq. 7. To compute $\Delta^{t+1,t}$ and the loss $L_{\text{fb}}$ of Eq. 6, we simply reverse the temporal order of the images in $\mathbf{P}^t$. The training of the two models only differs with respect to $L_{\text{det}}$. In the single view case, YOLOX is pretrained and kept frozen, and therefore $L_{\text{det}}$ is ignored. These losses are summed into the full training loss $L$ of Eq. 4 that we minimize over our training set to set the network weights $\theta$.

## 3.5 Inference and Track Creation

At inference time, we run our now trained network and obtain $\mathbf{X}^t$ and $\Delta^{t,t+1}$ for each time $t$.

### 3.5.1 Multi-view association

For the multiview model we perform non-maximum suppression in each $\mathbf{X}^t$ and use k-means clustering on the confidence of the remaining maximum (with $k = 2$) to separate *true detections* from noise. To link these detections into trajectories, we use MuSSP (Wang et al., 2019) that does this using a min-cost flow method that operates on a graph whose nodes are the detections connected by temporal edges that connect detections at different times. These edges are weighted as follows. An edge connecting locations $i$ and $j$ at times $t_1$ and $t_2$ respectively is taken to be

$$c\left(e_{i,j}^{t_1,t_2}\right) = -\frac{1}{e^{\sigma_t(|t_1-t_2|-1)}}\frac{1}{e^{\sigma_d d(i,j)}}W_m, \tag{8}$$

Where $d(i,j)$ is the Euclidean distance between the two locations $i$ and $j$. $\sigma_t$, $\sigma_d$ and $\sigma_m$ are hyperparameters that control the contribution of the temporal, spatial and motion-based distances between the detections, respectively. $W_m = \frac{1}{e^{\sigma_m d(i,j+|t_1-t_2|\delta_j^{t_2,t_2-1})}}$, is a new term used to model motion based distances between detections. Specifically $W_m$ measure the distance between location $i$ at time $t_1$ and the expected position of the detection at location $j$ after applying the estimated motion defined by $\delta_j^{t_2,t_2-1}$.

### 3.5.2 Single-view association

In our single-view pipeline, we retain the original post-processing step implemented by YOLOX. For track association, we modify ByteTrack (Zhang et al., 2022). First, we modify the detection association step to utilize ground plane IoU instead of image plane IoU. Secondly, we replace the Kalman filter used in ByteTrack for motion estimation with the learned predicted motion $\Delta^{t,t+1}$ output by our extended YOLOX.

## 4 Experiments

In this section, we empirically validate our approach. After defining the experimental protocol, we first evaluate the quality of the predicted motion. Subsequently, we assess the complete multi-view detection and tracking pipeline on the WILDTRACK dataset (Chavdarova et al., 2018), and the single-view pipeline on MOT17 (Milan et al., 2016). Finally, we conduct multiple ablations to explain the contribution of each component to the overall performance.

## 4.1 Experimental Setup

### 4.1.1 Datasets

We evaluate our approach on WILDTRACK (Chavdarova et al., 2018) and MOT17 (Milan et al., 2016).

**WILDTRACK**   is a calibrated multiview multi-person detection and tracking dataset featuring 7 viewpoints focusing on an area of $12 \times 36\text{m}^2$ in the real world. It contains 400 synchronized frames per view with a resolution of $1080 \times 1920$ pixels. Each person is annotated with a bounding box. As in Engilberge et al. (2023) we discretize the ground plane such that one cell corresponds to 20cm in the real world yielding a ground plane map of size $180 \times 80$ cells.

Table 1: **Tracking performance on WILDTRACK.** Our model significantly outperforms existing baselines across all tracking metrics. The performance values for MVDet and MVDeTr are taken from Engilberge et al. (2023), while those for other methods are sourced from their respective papers. For our method, we report both the best result and the average result over three runs.

| Model | WILDTRACK dataset | | | | |
|---|---|---|---|---|---|
| | MOTA | MOTP | IDF1 | IDP | IDR |
| DeepOcclusion+KSP [10] | 69.6 | - | 73.2 | 83.8 | 65.0 |
| DeepOcclusion+KSP+ptrack [10] | 72.2 | - | 78.4 | 84.4 | 73.1 |
| ReST [12] | 81.6 | - | 85.7 | - | - |
| MVDet [24] + muSSP | 80.6 | 0.80 | 79.4 | 79.2 | 79.6 |
| MVDeTr [23] + muSSP | 89.4 | 0.58 | 90.7 | 90.5 | 90.9 |
| EarlyBird [44] | 89.5 | - | 92.3 | - | - |
| MVFlow [16] | 91.3 | 0.57 | 93.5 | 92.7 | 94.2 |
| Ours mean±std | 91.7±0.2 | 0.56±0.02 | 93.9±0.2 | 93.0±0.2 | 94.9±0.1 |
| Ours best | **91.9** | **0.54** | **94.1** | **93.2** | **95.0** |

**MOT17** is a single-view, multi-person detection and tracking dataset comprising sequences captured by both static and moving cameras. The sequence lengths vary from 450 to 1500 frames, with frame rates ranging from 14 to 30 fps, totaling 15,948 frames. Video resolutions are either $1080{\times}1920$ pixels or $640{\times}480$ pixels, and each sequence contains up to 222 tracks. Every individual is annotated with a bounding box. For our method, we utilize the groundplane homographies $\mathbf{H}_c$ determined in Dendorfer et al. (2022) to map into the ground-plane. Since the test set is private we follow the train/val split of Zhou et al. (2020).

### 4.1.2 Implementation Details.

Our modes are implemented in Pytorch (Paszke et al., 2019) and trained on a single NVIDIA A100 GPU.

**Single-View Model** The detector is a modified YOLOX (Ge et al., 2021), as described in Section 3.4. We use the existing implementation of MMDetection (Chen et al., 2019) to develop our training pipeline. To augment the input images, we follow the strategy outlined in Ge et al. (2021). To accommodate pairs of frames as input, the detections are first extracted for the two frames using a frozen YOLOX, the detection are converted to gaussian heatmaps of 512 channels, first channel is a gaussian mask, while the remaining 512 channels are use to provide crop features of each detection. ResNet-50 (He et al., 2016) is used to extract feature for each heatmaps independently. The number of channels in the first layer of the ResNet is modified from 3 to 513 accordingly. The resulting features are concatenated along their channel dimension before being fed into two convolutional blocks consisting of ReLu(Nair & Hinton, 2010) - convolutional layer. The last convolutional layer output a 2 channels displacement map. The model is trained on MOT17 following the approach in Milan et al. (2016), with modifications as described in Section 3.4. Training images are supplied with random intervals between them to encourage the model to accurately learn motion behavior at various frame rates.

**Multi-View Model** The feature extractor is a ResNet-50 (He et al., 2016) whose last four layers have been removed such that it outputs feature maps containing 256 channels. The input images are augmented using a random crop strategy with a probability of 0.5. The scene aggregator is a multi-scale module, it comprises four scales. Between each scale the feature resolution is halved. Each scale consists of four blocks of "convolutional layer - batch normalization (Ioffe & Szegedy, 2015) - ReLu (Nair & Hinton, 2010)". The outputs of the four scales are bilinearly interpolated back to their original dimension. After concatenation, they go through a final $1{\times}1$ convolutional layer with 3 output channels. One channel goes through a sigmoid function to produce the detection probability map. The other 2 are taken to be the motion offsets. The model is trained on WILDTRACK using the Adam optimizer (Kingma & Ba, 2015) with a learning rate of 0.0001 and a batch size of one. The learning rate is halved after epochs 20, 40 and 60.

**Memory optimization** The differentiable reconstruction model of Section 3.2 aggregates the entire input detection map to reconstruct each location of the output. In practice this becomes too memory intensive

Table 2: **Tracking performance with low frame rate on MOT17 validation.** We also evaluate our approach in a monocular setting. We use the MOT17 dataset and modify YOLOX and Bytetrack to predict and use motion.

| | MOT17 val dataset | | | | |
|---|---|---|---|---|---|
| Model | FPS | Set | MOTA | MOTP | IDF1 |
| ByteTrack [52] | 0.75 | val | 51.1 | 84.6 | 55.0 |
| Ours | 0.75 | val | **58.9** | **84.8** | **58.3** |
| ByteTrack [52] | 2 | val | 59.1 | 84.4 | 59.6 |
| Ours | 2 | val | **65.5** | 84.4 | **62.9** |
| CenterTrack [55] | 30 | val | 61.1 | - | - |
| ByteTrack [52] | 30 | val | **77.0** | 84.1 | **77.5** |
| Ours | 30 | val | 76.6 | 84.1 | 76.1 |

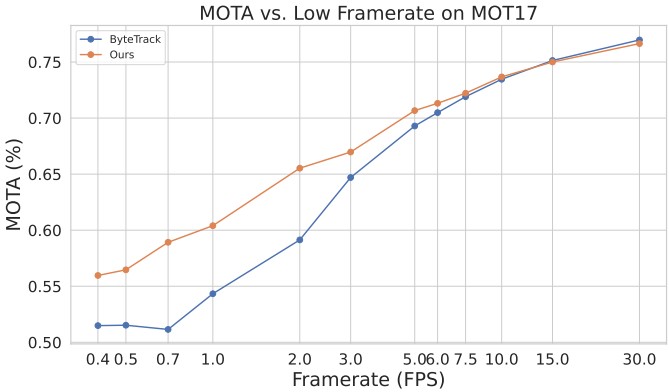

Figure 5: **Tracking results in low FPS scenario.** In low frame rate scenarios our model outperforms the ByteTrack baseline in term of MOTA. The lower the frame rate the higher the performance gap, showing the benefit of the proposed approach.

when the input/output resolutions increase. Therefore the reconstruction step is implemented with a sliding window, where the window size can be set to any value, making memory consumption fixed and not dependent on heatmap resolution. For all experiments we use a window size of 59 pixels, meaning that, at most the reconstruction step can handle motion offset of length $59/2 = 29.5$ pixels. Which is larger than any groundtruth motion.

**Hyperparameters** The reconstruction hyperparameter $\lambda_r$ is initialized to 0.8, during training it is increased at the end of every epoch by 0.08 until it caps out at 5. Additionally when computing the detection loss on reconstruction, the ground truth heatmap is also passed through the reconstruction module with a zero offset to have the same imprecision (scaled up Gaussian peak linked to the value of $\lambda_r$ ) on both the ground truth and prediction. The hyperparameters in the loss $L$ are set to $\lambda_{\text{fb}} = 0.05$ and $\lambda_{\text{se}} = 1$. A summary of the hyperparameters can be found in Table A.6.

### 4.1.3 Metrics.

To gauge the quality of the final trajectories, we report the same CLEAR MOT metrics (Kasturi et al., 2009) as in previous works Chavdarova et al. (2018); Engilberge et al. (2023). They include Multiple Object Detection Accuracy (MODA), Multiple Object Tracking Accuracy (MOTA), and Multiple Object Tracking Precision (MOTP). We also report identity preservation metric IDF1 (Ristani et al., 2016), computed from Identity Precision (IDP) and Identity Recall (IDR). Since we are concerned with motion and tracking people, MOTA, MOTP, and IDF1 are the most significant because they quantify the quality of trajectories, instead of that of detections in individual frames like MODA. We use the *py-motmetrics* and MMDetection (Chen et al., 2019) libraries to compute these metrics.

Since a specificity of our approach is to predict 2D motion offsets, we also seek to quantify how good they are. To this end, we compute the $L1$ distance between predicted and the ground truth offsets. For non-zero ground truth offsets, we also compute the average angular error and norm between ground truth and prediction. We will refer to these measures as L1, Angle. and Norm.

## 4.2 Comparative Results

### 4.2.1 Tracking evaluation

In Table 1, we compare the CLEAR MOT metrics (Kasturi et al., 2009) achieved by our method with those yielded by other state-of-the-art methods. On WILDTRACK, our approach significantly outperforms all others. For MOT17, results can be found in Table 2. Our approach surpasses the original YOLOX +

ByteTrack at most frame rates, thus demonstrating the advantage of learning to represent motion. In low frame rate scenarios, it significantly outperforms the baselines: at 2FPS, MOTA is improved by more than 10%. We further illustrate the ability of our single view method in low frame rate scenario in Fig. 5.

### 4.2.2   Motion evaluation

To evaluate, the quality of our motion offsets, we compare them to those produced by two additional baselines. First, since our motion offset representation shares similarities with optical flow, we used RAFT (Teed & Deng, 2020), a network pre-trained to estimate optical flow, to estimate the flow in each one of the viewpoints. The resulting flows are projected onto the ground plane and averaged. Second, we trained a version of our model using full supervision by skipping the reconstruction and instead directly applying an L2 loss between predicted and the ground-truth offset. In other words, unlike in our normal approach, we provide full motion supervision. The groundtruth motions are derived from identity annotations present in WILDTRACK. We report the results in Table 3. Our weakly supervised outperforms the RAFT baselines by significant margins and comes close to the fully supervised method on both datasets. In Fig. 6, we provide a qualitative result on WILDTRACK to show this visually. The supplementary material contains video results illustrating the robustness of our motion prediction method even in a low frame rate scenario.

Table 3: **Motion offset evaluation** We evaluate 2D motion offset according to three metrics, L1 error, Angular error (in degree) and Norm error. Our model outperforms the optical flow baseline by a large margin. It even performs competitively against the fully supervised methods, especially in terms of norm error.

|  | WILDTRACK dataset | | |
|---|---|---|---|
| model | L1 ↓ | Angle ↓ | Norm ↓ |
| RAFT [43] | 1.06 | 48.8 | 2.07 |
| Supervised | 0.55 | 32.5 | 0.90 |
| Ours | 0.58 | 34.8 | 0.92 |

Table 4: **Ablation results on WILDTRACK** We conduct an ablation study on three components of the system, the spatial extent term (Eq. (7)), the temporal consistency term (Eq. (6)) and the reconstruction step (Eq. (3)).

| Losses | | | Metrics | | |
|---|---|---|---|---|---|
| $L_{\text{mot}}$ | $L_{\text{se}}$ | $L_{\text{fb}}$ | MODA ↑ | L1 ↓ | Angle ↓ |
|  |  |  | 90.7 | 0.87 | 82.0 |
| ✓ |  |  | 89.6 | 0.59 | 36.2 |
| ✓ | ✓ |  | 90.0 | 0.65 | 43.8 |
| ✓ |  | ✓ | 91.3 | 0.59 | 35.0 |
| ✓ | ✓ | ✓ | 91.8 | 0.58 | 34.8 |

### 4.3   Ablation Study

To understand which components of the model contribute to the overall performance, we conduct an ablation study on the WILDTRACK dataset. We test three components of our model, the smoothing term $L_{\text{se}}$, the temporal consistency term $L_{\text{fb}}$ of our loss both defined in Section 3.3 and the reconstruction from motion defined in Section 3.2. To disable the reconstruction from motion, smoothing and temporal consistency we simply omit the losses.

As it can be seen in Table 4, adding $L_{\text{mot}}$ greatly improves the motion metrics at the cost of slightly degrading the MODA one. This was to be expected since the same model has to predict motion on top of the original detection. Further adding $L_{\text{se}}$ and $L_{\text{fb}}$ only has beneficial effect both in terms of MODA, L1, and angle error. The best score is obtained when the three elements are combined.

We also conduct an ablation study on the benefit of using motion offset in the muSSP (Wang et al., 2019) association algorithm. We modify Eq. (8) by removing the last term $W_m$ and hence ignore the predicted motion offset. When muSSP is executed without motion offset MOTA drops by 0.5 points and IDF1 0.9 by points compared to the last row of Table 1. This confirms the usefulness of integrating motion offset inside association algorithms.

### 4.4   Limitations

One limitation is tied to the formulation of $L_{mot}$. It has multiple local minima, any motion flow that correctly maps the detection from one timestep to the next would yield a zero loss. In practice and when combined

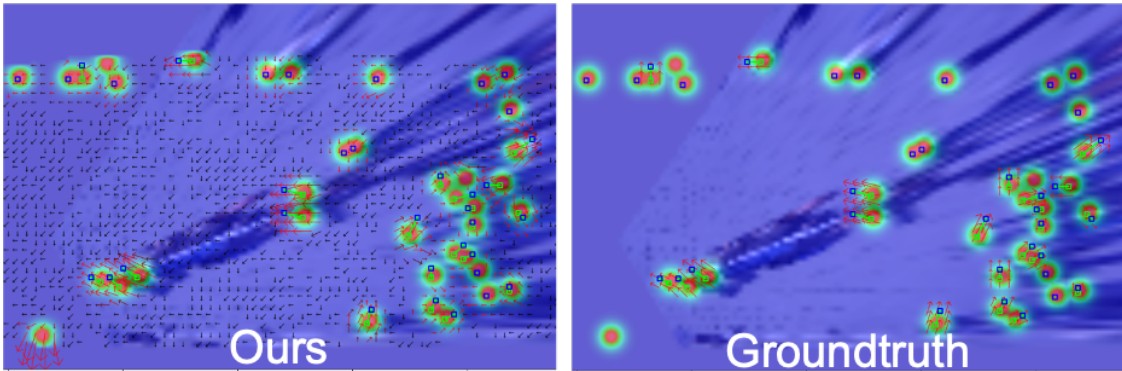

Figure 6: **Visualizing the Motion Offsets.** We visualize the detection and the motion offset predicted by our model on the left and the ground truth on the right. The green squares mark the position of the person at time $t$ and the blue square the position at time $t+1$. The green arrows connecting them denote the ground truth motion. The offset map is visualized by regularly drawing arrows corresponding to the motion offset. Qualitatively the predicted detection and motion offset are very close to the ground truth ones. The quantitative results reported in Table 3 confirm this.

with $L_{se}$ and $L_{fb}$ we observe that the model converges to predicting the real motion. Additional results are provided in the appendix to showcase this.

A second limitation of the proposed differentiable reconstruction from motion is its memory cost. The memory consumption grows quadratically with respect to the dimension of the space it is reconstructing. To mitigate this difficulty. the implementation relies on a sliding window mechanism, as described in Section 4.1. This fixes the required amount of memory but also limits the maximum range of motion to half the size of the window. In practice it hasn't been a problem, but it could be one for very high resolution scenarios that feature long-range motions. In such a scenario memory would be a bottleneck anyway, and parallelizing the workload across multiple GPUs would be the way to overcome this limitation.

## 5 Broader Impact

Our research contributes to the field of multi-object tracking by proposing a motion prediction method that reduces the need for costly identity annotations, relying only on detection supervision. This development has significant implications for real-world applications such as surveillance, autonomous driving, and public safety, where robust tracking is crucial but manually labeling motion data is prohibitively expensive. By improving tracking performance in challenging low-frame-rate scenarios and crowded environments, this work has the potential to enhance the efficiency and accuracy of systems that monitor human activity, making them more accessible and scalable.

However, the deployment of such technologies must be done with care, as they could raise privacy concerns if used for mass surveillance or other invasive purposes. It is important that future applications of this work consider ethical implications and adhere to strict guidelines to ensure the protection of individual privacy and mitigate risks of misuse.

## 6 Conclusion

We have proposed a network to predict both detection heatmaps and motion offsets given only detection supervision. To this end, we use the consistency between the predicted heatmaps and offsets as a supervisory signal. We have demonstrated that this increases performance in challenging single- and multi-view people-tracking scenarios. The effect of this approach is particularly noticeable in low frame-rate scenarios, where conventional tracking-by-detection approaches are most likely to fail.

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

## Appendix

Appendix is organized as follows: Appendix A provides a detailed evaluation of the predicted displacement and additionnal analysis of the single-view model. Appendix B offers additional quantitative results on MOT17. In Appendix E, we present visualizations of the motion offsets. Appendix F details tracking results for an additional dataset, MultiviewX. The supplementary archive also contains a discussion about broader impact of the work Section 5, video examples discussed in Appendix G and the code described in Appendix H.

## A Further Analysis of Single-View Model

### A.1 Comparison with Kalman Filter

The original ByteTrack algorithm uses a Kalman Filter to estimate the motion of objects between frames. We aim to assess the contribution of the Kalman Filter to ByteTrack's performance and compare it to our proposed motion estimation method.

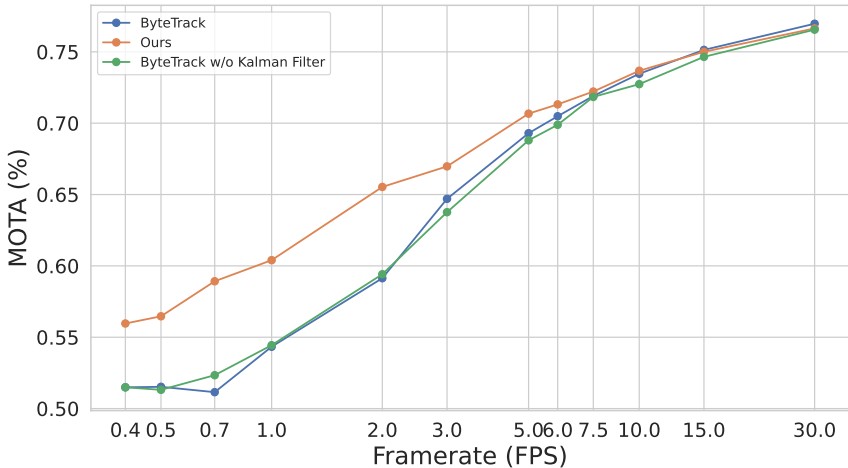

Figure A.1: **Comparison with Kalman Filter:** In our adaptation of ByteTrack, we substitute the Kalman Filter with the motion offset predicted by our method. This comparison demonstrates that the Kalman Filter exerts a negligible influence on ByteTrack, offering slight enhancements at high frame rates while providing no benefit or even impairing performance at lower frame rates. In contrast, our approach markedly enhances performance at low frame rates.

To this end, we present results at various frame rates for a modified version of ByteTrack where the Kalman Filter has been removed. In other words, new detections are directly matched to the head of existing trajectories.

The results of this modified model can be found in Figure Fig. A.1. Surprisingly, the motion estimated through the Kalman Filter only has a marginal effect on the tracking performance. At high frame rates (> 7.5 FPS), MOTA slightly increases, while at lower frame rates, it has no effect or is detrimental. Our weakly supervised method for motion estimation performs significantly better than the Kalman Filters.

### A.2 Additional Motion Baselines

As discussed in the limitations section of the main article, the proposed method for training a motion predictor with detection supervision can have multiple global minima. Any displacement map that maps all motion at timestep $t$ to a unique detection at timestep $t + 1$ would result in $L_{\mathrm{mot}} = 0$. In practice, when combined with the smoothing term $L_{se}$ and the time consistency term $L_{fb}$, we observe that the model

converges to predicting the actual motion. We argue that, considering time consistency and given the provided data, it is easier for the model to learn the true motion than any other solution.

While the results in Table 3 of the main paper already demonstrate the quality of the predicted displacements, we propose to compare the motion estimated by our model to two other baselines that could minimize the loss $L_{\mathrm{mot}} = 0$.

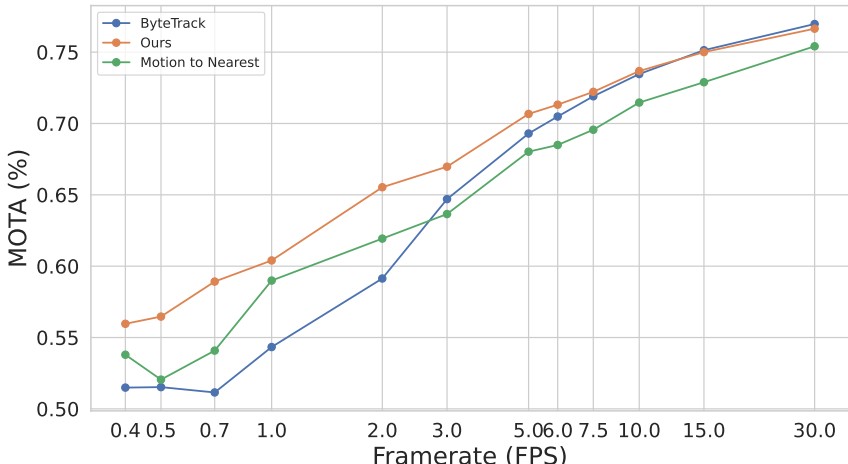

Figure A.2: **A Simple Motion Baseline:** We compare our motion prediction to a basic baseline where the motion from one timestep to the next is estimated by identifying the nearest detection. Utilizing this motion in place of the Kalman filter in ByteTrack results in the green curve depicted above. Surprisingly, this straightforward baseline outperforms ByteTrack at low FPS; however, it is disadvantageous at high FPS. Our methods outperforms this baseline in both scenarios.

### A.2.1 Motion to Nearest

First, we will test a baseline that estimates motion such that it maps a detection at time $t$ to the nearest detection at time $t + 1$. The results for this baseline can be found in Fig. A.2.

Surprisingly, such a simple baseline outperforms ByteTrack at low frame rate regimes (<3 FPS) and underperforms at higher FPS. When comparing the proposed motion estimation method to this baseline, the weakly supervised motion greatly outperforms the baseline. This would tend to confirm that the proposed model is not merely learning to map detections to their nearest neighbors.

### A.2.2 Motion from Bipartite Matching

One could argue that the previous baseline does not guarantee $L_{\mathrm{mot}}$ to be zero, since multiple detections can be mapped towards the same detection, which is indeed a valid concern. Therefore, we propose a second baseline that guarantees reaching a global minimum of $L_{\mathrm{mot}}$. We suggest estimating motion using a bipartite matching method to match detections from time $t$ with those from time $t + 1$, utilizing the Hungarian algorithm.

The results can be found in Fig. A.3. Contrary to expectations, the bipartite matching baseline performs worse than the nearest neighbor approach. This is likely due to the one-to-one matching constraint, which, when applied to noisy detections, can lead to detections being wrongly matched with distant ones, resulting in large motion errors. Even adding a cutoff distance does not significantly improve performance. Nonetheless, the bipartite matching baseline is slightly beneficial in very low frame rate scenarios (<2 FPS) but detrimental otherwise. When compared to the proposed approach, its performance is significantly lower, confirming that the proposed model is not learning a form of bipartite matching but is more likely capturing the true motion.

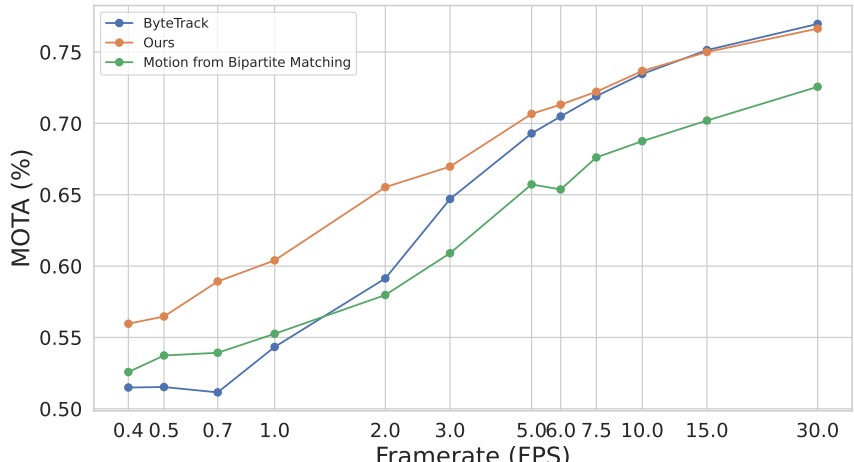

Figure A.3: **Motion from Bipartite Matching:** ByteTrack employs the Hungarian algorithm to associate detections between timesteps using a similarity score computed via the Intersection over Union (IoU) of bounding boxes. This approach fails when the motion is significant and the intersection is null. To address this, we propose a straightforward baseline utilizing the Hungarian algorithm to estimate the motion between detections at two timesteps. Similar to the previous method, this estimated motion replaces the Kalman filter in ByteTrack. The results, represented by the green curve, indicate a marginal benefit at very low frame rates ($<2$ FPS) and are otherwise detrimental.

Additional qualitative results for both baselines can be found in Fig. A.7.

### A.3   Tracking on the Groundplane

Our method utilizes the ground plane to predict motion and associate detections at low frame rates. To assess its contribution to the overall performance of our model, we propose comparing it with a ground plane-adapted version of ByteTrack.

The association step in ByteTrack is modified as follows: Detections are projected onto the ground plane: The bottom center of the bounding box is projected onto the ground plane using the frame homography $H$. This ground point serves as the center of a square ground bounding box, with a side length of 25. These ground bounding boxes are used to compute similarity scores, which replace ByteTrack's original similarity scores. Results for this baseline can be found in Fig. A.4.

Below 5 FPS, ByteTrack Ground significantly outperforms ByteTrack, which is expected since operating on the ground plane mitigates the effects of perspective projection, thereby making motion and speed uniform across the scene and simplifying association. Above 5 FPS, it performs worse than ByteTrack, which may be attributed to noise in the detections and inaccuracies in ground plane projection.

Overall, our method significantly outperforms the ByteTrack Ground baseline across all frame rates. The performance gap between the two reflects the superiority of our motion estimation.

### A.4   Detection Features

In the single-view scenario, our motion prediction model is trained solely using the detection map. To enrich the scene representation, the input map can be augmented by appending convolutional features corresponding to each detection onto their channel dimension. To achieve this, we utilize a pretrained OSNet (53) to extract features with a dimensionality of 512 for each detection bounding box. It's important to note that these features are solely provided as input to the model and are not utilized to measure the similarity between detections.

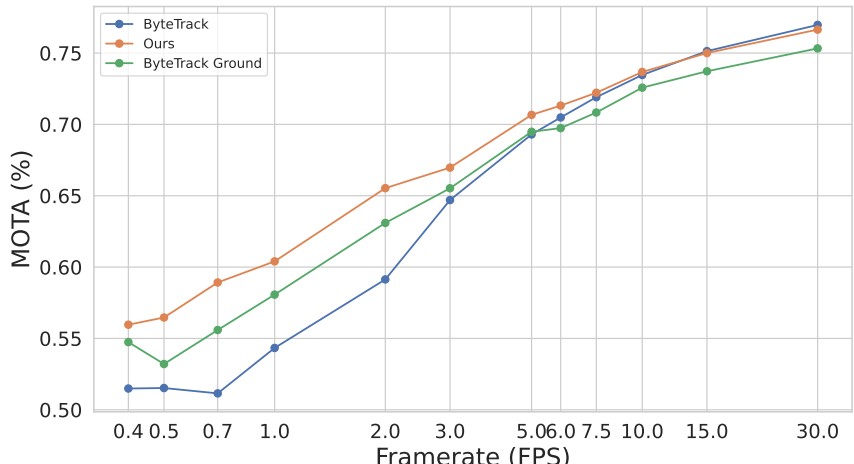

Figure A.4: **Effect of Tracking on the Ground Plane:** Working on the ground plane significantly simplify tracking. In the image plane, individuals appear to move faster when closer to the camera due to perspective effects. By negating this perspective effect, trajectories across the scene become more uniform. This uniformity is especially advantageous at low frame rates, where motion is more pronounced. However, at higher frame rates, this approach can be slightly counterproductive, likely due to calibration inaccuracies or noise in the detection process.

As demonstrated in Table A.1, the inclusion of detection features, regardless of the frame rate, enhances the tracking performance of our motion prediction network.

## A.5 Time Interval

Given that our model processes pairs of frames and aims to predict accurate motion across a broad range of frame rates, we explore different strategies for sampling training pairs. In Table A.1, we present results from training pairs sampled with intervals ranging from $[1 - 2[$ to $[8 - 13[$. These findings indicate that intervals more closely aligned with the target frame rate lead to superior outcomes. Consequently, for all our experiments, we select the training sampling interval based on the desired target frame rate.

Table A.1: **Ablation Results on MOT17:** We evaluate two additional elements of our training process: the interval between sampled frame pairs during training and the inclusion of detection features in the input map. Incorporating detection features appears advantageous across all scenarios. Moreover, the training frame interval achieves optimal results when it closely mirrors the inference regime. A frame interval of 1 corresponds to 30 FPS, whereas an interval of 10 equates to 3 FPS.

| Model component | | FPS | Metrics | | |
|---|---|---|---|---|---|
| Frame interval | Detetection features | | MOTA ↑ | MOTP ↑ | IDF1 ↑ |
| $[1 - 2[$ | | 2 | 59.4 | 84.3 | 60.6 |
| $[1 - 2[$ | ✓ | 2 | 61.3 | 84.6 | 61.9 |
| $[8 - 13[$ | ✓ | 2 | 65.5 | 84.3 | 58.0 |
| $[8 - 13[$ | | 30 | 75.7 | 84.1 | 73.2 |
| $[8 - 13[$ | ✓ | 30 | 76.6 | 84.1 | 76.1 |
| $[1 - 2[$ | | 30 | 76.6 | 84.1 | 76.0 |
| $[1 - 2[$ | ✓ | 30 | 76.9 | 84.1 | 77.6 |

## A.6  Faster R-CNN detector

Since all our single-view results use a pre-trained YOLOx detector, we propose to evaluate our approach with a less performant detector: Faster R-CNN (36).

Table A.2: **Tracking performance with Faster-RCNN detector.** We also evaluate our approach in a monocular when combined with a two stage detector Faster-RCNN. Best results with Faster R-CNN detector are underlined

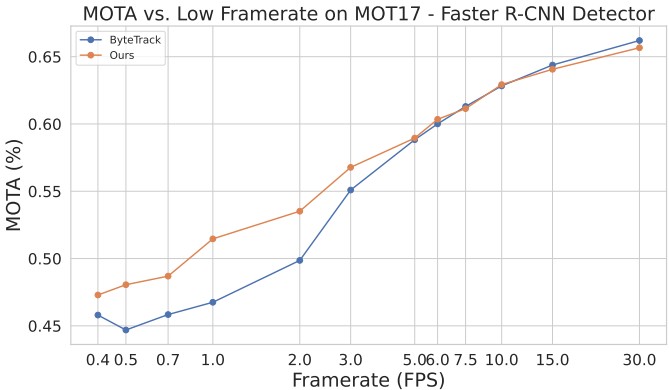

Figure A.5: **Tracking results in low FPS scenario.** When using a Faster-RCNN object detector the proposed approach outperform the Bytetack approach. The lower the frame rate the higher the performance gap, showing the benefit of the proposed approach.

|  | | MOT17 val dataset | | |
| --- | --- | --- | --- | --- | --- |
| Model | FPS | Set | MOTA | MOTP | IDF1 |
| ByteTrack [52] | 0.75 | val | 51.1 | 84.6 | 55.0 |
| Ours | 0.75 | val | **58.9** | **84.8** | **58.3** |
| ByteTrack [52] + FRCNN | 0.75 | val | 45.8 | 80.9 | 51.2 |
| Ours + FRCNN | 0.75 | val | 48.7 | 80.9 | 53.0 |
| ByteTrack [52] | 2 | val | 59.1 | **84.4** | 59.6 |
| Ours | 2 | val | **65.5** | **84.4** | **62.9** |
| ByteTrack [52] + FRCNN | 2 | val | 50.0 | 80.6 | 54.6 |
| Ours + FRCNN | 2 | val | 53.5 | 80.5 | 67.8 |
| ByteTrack [52] | 30 | val | **77.0** | **84.1** | **77.5** |
| Ours | 30 | val | 76.6 | **84.1** | 76.1 |
| ByteTrack [52] + FRCNN | 30 | val | 66.2 | 80.4 | 71.0 |
| Ours + FRCNN | 30 | val | 65.7 | 80.4 | 67.8 |

**Quantitative results**   The results are shown in Table A.2 and Fig. A.5. As expected, both ByteTrack + Faster R-CNN and Our + Faster R-CNN perform worse than their counterparts using YOLOx. Nonetheless, our approach, even when combined with Faster R-CNN, surpasses ByteTrack + Faster R-CNN at most frame rates. These results are consistent with those obtained using the YOLOx detector and demonstrate that the proposed approach performs reliably across different object detectors.

## A.7  Memory Consumption and Runtime

In addition to tracking performance, we provide an analysis of the memory usage of our approach for both training and inference on the MOT17 dataset. Results can be found in Table A.3

During training, the memory consumption of the differentiable reconstruction step grows quadratically with respect to the dimension of the space being reconstructed. In our implementation, this is managed using a sliding window mechanism. During inference, the reconstruction step is not needed, and the memory overhead of our approach is minimal.

Table A.3: **Memory Consumption** Training and inference GPU memory consumption for ByteTrack (YOLOx) and our method (motion prediction). Tests were run on an NVIDIA V100 with a batch size of 1 during inference and 2 during training. When training our method, YOLOx is kept frozen, so its memory footprint is not included. During inference, the memory consumption of ByteTrack (YOLOx) is added to the memory footprint of our method.

| Model | Phase | Memory |
| --- | --- | --- |
| ByteTrack [52] | Training | 29.4 GiB |
| ByteTrack [52] | Inference | 1.7 GiB |
| Ours | Training | 5.2 GiB |
| Ours | Inference | 1.7 + 0.7 GiB |

# B    Additional MOT17 results

Due to the private nature of the test set and the need to conduct experiments at various frame rates, results in the main paper are conducted exclusively on a validation set, obtained by splitting the original training data in half. The first half is used for training and the second one for validation. For more information about the data split, see (52).

For completeness, we provide results at 30FPS on the private test set. Results can be found in Table A.4. The results on the test set are consistent with those on the validation set and are on par with the original ByteTrack (52). Note that at 30FPS, the overlap of consecutive objects in a trajectory is large and the motion is small. In such scenarios, there is little benefit in using motion; nonetheless, our method is able to preserve the original performance.

Table A.4: **Tracking performance on MOT17 test set with private detectors.** At high FPS, predicting motion is of limited interest, nonetheless our method obtains results on par with the original ByteTrack.

| | MOT17 val dataset | | | | |
|---|---|---|---|---|---|
| Model | FPS | Set | MOTA | MOTP | IDF1 |
| CenterTrack [55] | 30 | test | 67.8 | - | 64.7 |
| ByteTrack [52][1] | 30 | test | **77.1** | 80.7 | **74.4** |
| Ours | 30 | test | 76.7 | **80.8** | 72.4 |

# C    MOT 20 Results

We provide results for our single view model on the MOT20 datasets.

Table A.5: **Tracking performance with low frame rate on MOT20 validation.** We also evaluate our approach in a monocular setting. We use the MOT20 dataset and modify YOLOX and Bytetrack to predict and use motion.

| | MOT20 val dataset | | | | |
|---|---|---|---|---|---|
| Model | FPS | Set | MOTA | MOTP | IDF1 |
| ByteTrack [52] | 0.75 | val | 23.1 | **79.0** | 24.9 |
| Ours | 0.75 | val | **30.9** | **79.0** | **25.0** |
| ByteTrack [52] | 2 | val | 51.2 | **78.8** | **45.9** |
| Ours | 2 | val | **55.5** | **78.8** | 45.8 |
| ByteTrack [52] | 30 | val | **70.7** | **78.6** | **70.5** |
| Ours | 30 | val | 70.6 | **78.6** | 69.8 |

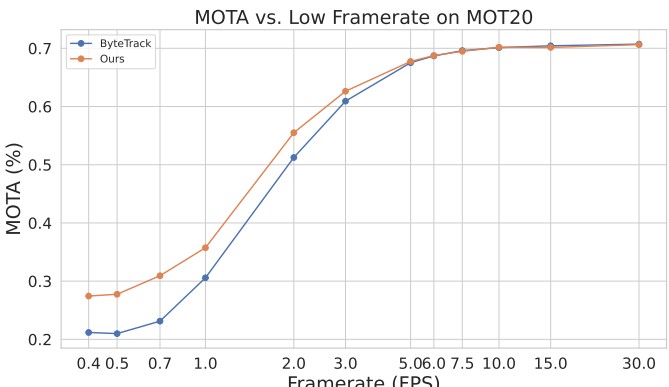

Figure A.6: **Tracking results in low FPS scenario.** In low frame rate scenarios our model outperforms the Byte-Track baseline in term of MOTA. The lower the frame rate the higher the performance gap, showing the benefit of the proposed approach.

**MOT20**  is a single-view, multi-person detection and tracking dataset comprising sequences captured by both static and moving cameras. The sequence lengths vary from 429 to 3315 frames, with frame rates of 25 fps, totaling 13410 frames. The sequence represent busy scene and can contains up to 1211 tracks. Every individual is annotated with a bounding box. For our method, we utilize the groundplane homographies $\mathbf{H}_c$

---

[1]Results obtained using MMDetection implementation of ByteTrack

determined in Dendorfer et al. (14) to map into the ground-plane. Since the test set is private we follow the train/val split of MMDetection.

**Quantitative results**    Results can be found in Table A.5 and Fig. A.6. Our approach surpasses the original YOLOX + ByteTrack at most frame rates, thus demonstrating the advantage of learning to represent motion. In low frame rate scenarios, it significantly outperforms the baselines: at 2FPS, MOTA is improved by more than 10%. Those results are in line with the results on MOT17 and show that the proposed approach performs consistently across datasets with various crowd densities.

# D    Hyperparameter Details

In this section, we provide details about the hyperparameters used in our approach. These hyperparameters were carefully selected through experimentation to achieve the best performance.

Table A.6: **Hyperparameters and their default values**

| Hyperparameter | Description | Single View | Multi View |
|---|---|---|---|
| Learning rate | Adam learning rate | 0.001 | 0.0001 |
| $\lambda_r$ | Trade off reconstruction accuracy/differentiability | 0.8 | 0.8 |
| $\lambda_{fb}$ | Weight of forward backward loss | 0.05 | 0.05 |
| $\lambda_{se}$ | Weight of Spatial extent loss | 1 | 1 |

The table in Table A.6 lists the hyperparameters along with their corresponding values. Note that for most hyperparameters the value is the same in the single and multiview use cases, it is also kept the same across datasets.

# E    Qualitative Results

We present qualitative results for various scenes from MOT17 in Fig. A.7. When compared with the two aforementioned baselines, motion-to-nearest detection and bipartite matching, our proposed method more accurately approximates the ground truth motion, despite being trained solely with detection supervision.

# F    MultiviewX Results

## F.1    Motion Offsets

We also evaluate the predicted motion offsets on the MultiviewX dataset. Results are presented in Table A.7.

Table A.7: **Motion offset evaluation** We evaluate 2D motion offset according to three metrics, L1 error, Angular error (in degree) and Norm error. Our model outperforms the optical flow baseline by a large margin. It even performs competitively against the fully supervised methods, especially in terms of norm error.

| | MultiviewX dataset | | |
|---|---|---|---|
| model | L1 | Angle | Norm |
| RAFT [43] | 2.56 | 50.3 | 2.84 |
| Supervised | 1.26 | 12.6 | 1.58 |
| Ours | 1.44 | 19.4 | 1.59 |

Table A.8:    **Tracking performance on MultiviewX.** Our model outperforms existing baseline on the metrics related to trajectories by a significant margin. Performance numbers for MVFlow (16) was generated from publicly available code.

| | MultiviewX dataset | | | | |
|---|---|---|---|---|---|
| Model | MOTA | MOTP | IDF1 | IDP | IDR |
| MVFlow [16] | 66.2 | 0.87 | 50.2 | **56.6** | 45.1 |
| Ours | **81.1** | **0.55** | **55.1** | 54.4 | **55.8** |

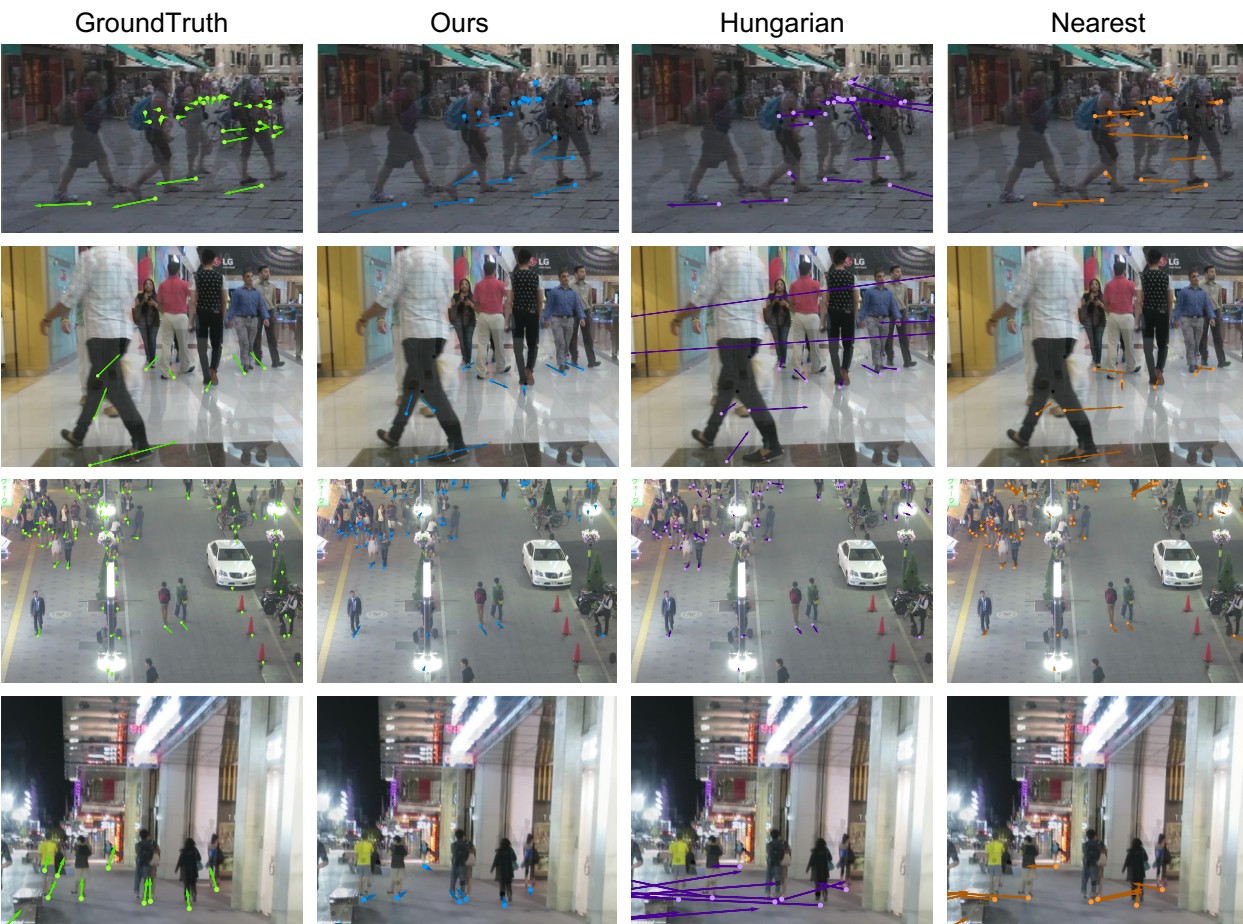

Figure A.7: **Qualitative Results on MOT17:** We compare our motion prediction model against two baseline methods: the Nearest Neighbor and the Hungarian algorithm. In both cases, the motions predicted by our model more closely align with the ground truth. Best viewed when zoomed in.

## F.2 Tracking

Previous methods have utilized the MultiviewX dataset solely for evaluating multiview detection tasks. We extend its use to include tracking evaluation. Employing the same evaluation protocol as in the WILD-TRACK dataset, we report our findings in Table A.8. Utilizing publicly available code, we establish a baseline for the MVFlow method (16). Our model demonstrates a significant improvement over MVFlow on the MultiviewX dataset.

## G  Video Results

The supplementary archive illustrate our model's capability in motion prediction, relying solely on detection data for supervision. Included is the video wild_offset.mp4, highlighting our model's effectiveness on the WILDTRACK dataset. The video features detection and offset maps projected onto the ground plane from the perspective of camera 1.

### G.1  Low frame rate Results

The video wild_extended_low_fps.mp4 illustrates our model's capabilities in a low frame rate setting. We augmented the original WILDTRACK dataset's frame rate from 2 fps to 10 fps by adding frames through

interpolation and adjusting the ground truth annotations accordingly. A specialized model was trained on this enriched dataset. For both the training and evaluation phases, we used frame pairs with a 15-frame interval, yielding an effective frame rate of $\frac{2}{3}$ fps. Our model demonstrates robust motion prediction capabilities even at this reduced frame rate.

## H   Code

We provide the PyTorch implementation of the differentiable reconstruction step in our approach. The code to reproduce some of the experiments is available at `https://github.com/cvlab-epfl/noid-nopb`.

