# OpenReview forum: "No Identity, no problem: Motion through detection for people tracking"
_TMLR — Accepted by TMLR_

### Review · Reviewer_Nt6h · 2024-08-28

**Summary Of Contributions:**

The paper proposes a method for predicting object motion using only detected objects as a supervisory signal. This eliminates the need for motion and identity annotations that are costly and hard to find for existing data. The method involves generating detection heat maps at two different time steps along with a 2D motion estimate between the two time steps. The motion estimate is then used to warp one heat map to match the other, and the resulting consistency between the two heat maps acts as the supervisory signal.

The paper introduces the method and conducts a series of experiments on the MOT17 and WILDTRACK datasets. These allow for experiments on both single- and multi-view inputs, and the dual-pathway network can accommodate both. The results show that this method outperforms state-of-the-art methods particularly in settings with low frame rates, constituting another data limitation that this method can help mitigate.

**Audience:**

Yes

**Claims And Evidence:**

Yes

**Requested Changes:**

Minor
- Some more ablations as described above, and some more results to quantify the usability of this system for those who need it
- Experiments with poorer object detectors (a small set)

**Strengths And Weaknesses:**

### Strengths
- Supervision approach is very clever - warping between object detection heat maps at two time steps is an innovative way to take advantage of video data and powerful object detection algorithms to create a supervisory signal for the goal, motion prediction
- Loss functions and architecture are described clearly. Overall, the method is pretty well-communicated
- Experimental setup is well-designed and supports the contributions of the paper
- Impact is convincing
  - Improved results on crowded scenes and low frame rates are demonstrated
  - Ability to take advantage of existing data that doesn't have complex ground-truth annotations without paying extra costs to collect annotations

### Weaknesses
- System requires high-quality initial detections. While we do have good tools for object detection, it would be interesting to see how this fares under various conditions where object detection is harder (not just crowds, for example), or where less performant detectors are being used
- The approach is somewhat complex beyond just loss, particularly with elements like the dual-pathway architecture. While the ablations on different parts of the loss are worthwhile, they could be extended to other aspects beyond loss.
- Memory intensiveness could use more concrete documentation and quantification. The efficacy in weak data regimes is a main selling point, so while memory intensiveness is not the same as needing costly annotations, potential users might benefit from the budget prediction.

---

> ### Author Response · Authors · 2024-09-24
> **Providing Additional Analysis: Response to Review**
>
> ### Faster R-CNN Results
> - In the appendix, we have included additional results using a different detector: Faster R-CNN. While the overall performance is lower compared to YOLOx, we observe similar trends. On MOT17, when comparing ByteTrack and our approach, both using the Faster R-CNN detector, our model outperforms ByteTrack in low-framerate scenarios.
>
> ### Further Ablations
> - Although Figures 4 may seem complex, our approach builds on common architectural choices. Nonetheless, the appendix includes several analyses of our single-view model to justify some of those choices. Specifically, we evaluate the impact of the time interval used when sampling image pairs and the use of CNN features to represent bounding boxes in motion prediction.
>
> ### Memory Usage
> - Regarding memory consumption, we would like to clarify that the memory-intensive component of our approach is the differentiable reconstruction step, which is only required during training. During inference, the reconstruction from motion offsets is not necessary, as the offsets can be used directly. As a result, the memory footprint at inference is comparable to that of the detector alone. Nonetheless, we have added a quantification of memory usage in various scenarios in the appendix, Section A.7.
>
> ### Broader Impact
> - A broader impact section has been included in the manuscript, section 5.

---

### Review · Reviewer_XLcJ · 2024-08-30

**Summary Of Contributions:**

The authors propose to use a detection model to solve multi-target tracking tasks. The key idea is to predict the offset between a pair of image frames to associate the targets presented on both images. The model can reconstruct the object's position at future frames with the detection results and the offset map predicted between frames. The paper's key contribution is to construct such a method upon existing detectors and demonstrate the effectiveness of the proposed paradigm. A set of corresponding implementations, including loss functions, network architecture, and experiments are presented. The results on single-view and multi-view multi-object tracking datasets are provided to help understand the significance of the proposed method.

**Audience:**

No

**Broader Impact Concerns:**

Multi-object tracking datasets involve human face, which is a long-standing legacy issue in this community. Such potential privacy issues should be paid attention to and discused.

**Claims And Evidence:**

No

**Requested Changes:**

1. To demonstrate the effectiveness of the proposed method, experiments on standard multi-object tracking benchmarks are required. To have strong evidence about the performance for tracking in `crowded scenes`, I would recommend adding the experiments on DanceTrack and MOT20 datasets. Previous methods, such as ByteTrack [Zhang et al. 2022] and OC-SORT[3] have provided publicly available YOLOX detector models and weights on these datasets so it would be easy to achieve fair comparison by reusing the detector.
2. The splitting of MOT17 train-val sets introduces potentially biased results. I suggest adding experiments on MOT17-test set.
3. The difference of the proposed method and the long-standing CenterTrack should be explicitly discussed. Given the very similar idea and practice, the experiments on related datasets can be added, especially considering that they bring extra benefit of demonstrating the performance under low frame-rate cases.


Reference:
[3] Cao, Jinkun, et al. "Observation-centric sort: Rethinking sort for robust multi-object tracking." Proceedings of the IEEE/CVF conference on computer vision and pattern recognition. 2023.

**Strengths And Weaknesses:**

Strengths:
1. The proposed method follows an intuitive and practically convincing path to using the detection models for multi-object tracking.
2. The authors present the results in both single-view and multi-view settings, which are helpful for broader use of the community.


1. The claim "In this paper, we propose exploiting motion clues while providing supervision only for the detections, which is much easier to do." can be potentially misleading. Given two frames with multiple targets presented, the identity of objects is required to associate the ground truths on time step t and t+1. In this sense, not just "supervision only for the detections" is required because only bounding boxes without identities are not enough to train the model.
2.  The claim "In this manner, we couple the information obtained from different images during training and increase accuracy, especially in crowded scenes and when using low frame-rate sequences." may need more evidence to be supported. To be precise, the experiments to demonstrate the effectiveness of the proposed method are conducted on WILDTRACK and MOT17. Neither of them is usually considered as "crowded scene". In contrast, in the community of multi-object tracking, some other datasets are used typically to demonstrate the effectiveness of tracking in crowded scenes, such as MOT20[1] and DanceTrack[2].
3. Though novelty is not a main concern when considering a TMLR submission, this paper's idea is very much similar to CenterTrack [Zhou et al. 2020]. CenterTrack also provides experiments on KITTI and Nuscenes, which are helpful to be added especially considering that they are better fit to study multi-object tracking in `low frame-rate sequences` cases.
4. The splitting of MOT17 train-val, following CenterTrack [Zhou et al. 2020], has been extensively discussed and considered biased because the training and validation subsets are from the same set of videos. I would encourage the authors to test the results on the MOT17-test set by submitting them to the evaluation server. Moreover, this raises a concern about overfitting: Compared to ByteTrack, which does not use learnable modules for association, the trained modules in the proposed method are more likely to overfit MOT17-train and achieve an unfair advantage when evaluating on MOT17-val. Given the limited baseline methods used for MOT17 experiments, the results on this benchmark are not convincing to me.

Reference:
[1] https://motchallenge.net/data/MOT20/
[2] Sun, Peize, et al. "Dancetrack: Multi-object tracking in uniform appearance and diverse motion." Proceedings of the IEEE/CVF Conference on Computer Vision and Pattern Recognition. 2022.

---

> ### Author Response · Authors · 2024-09-24
> **Clarifying Motion Supervision and Benchmark Choices: Response to Review**
>
> ### Clarification of Contribution
> - The comment, "Given two frames with multiple targets presented, the identity of objects is required to associate the ground truths on time step t and t+1," suggests a misunderstanding of our contribution. Our approach does not rely on the identity of objects for frame-to-frame association. As outlined in Section 3.2, none of the losses used to train our model depend on ground truth motion offsets. Instead, detection supervision serves as a weak signal for motion supervision, which is achieved through the proposed differentiable reconstruction mechanism described in Section 3.2.
>
> ### Distinction from CenterTrack
> - Our method differs significantly from CenterTrack. Although both approaches represent motion as a 2D offset, which is a standard representation, our method does not require any external supervision of the motion offset, whereas CenterTrack does. As stated in our response to Reviewer 81nZ, this allows us to get rid of "Motion Annotations," which are expensive to obtain. Thus, our approach represents an important step forward. Additionally, CenterTrack is designed as an all-in-one tracker. While we demonstrate our approach in a tracking setup, our primary focus is on predicting motion without direct supervision. As shown in our experiments, the proposed method can be integrated with existing trackers across various setups to enhance their performance.
>
> ### Comment on Benchmark Choice
> - We hope the above clarifications help justify our choice of experiments and benchmarks. While achieving state-of-the-art results on every tracking benchmark is commendable, our primary goal is not to outperform every model. Instead, we focus on demonstrating the efficacy and utility of our motion prediction approach. Thus, our evaluation consists of two parts: in Section 4.2.1, we show that when combined with existing trackers, our method improves tracking accuracy in both single and multi-view scenarios. In Section 4.2.2, we directly evaluate the predicted motion offsets themselves.
>
> ### Comments on Train/Val Split
> - We acknowledge that the train/val split used in the MOT17 experiments may not be ideal. However, due to the private nature of the MOT17 test set, this was our only available option for evaluating our model at lower frame-rates. Additionally, we have provided results on the MOT17 test set in the appendix (Section B) at 30 FPS, which are consistent with the results obtained on the validation set. As both models were trained on the exact same data, we believe the comparison to be as fair as possible.
>
> - Regarding the comment on overfitting, we disagree that any overfitting would provide an unfair advantage to our method. Bytetrack’s association process relies on YOLOx predictions, including bounding boxes, their confidence scores, and potentially re-identification features, all of which would also be impacted by overfitting.
>
> ### MOT20
> - As requested, we have included results for the MOT20 dataset in the appendix (Section C). These results exhibit a similar trend to those on MOT17, further confirming the advantage of our method in low-framerate scenarios, regardless of crowd density.
>
> ### Broader Impact
> - A broader impact section has been included in the manuscript, section 5.

---

> ### Comment · Reviewer_XLcJ · 2024-10-20
> **Thanks for the clarifictaion and additional experimental evidence**
>
> I appreciate the clarification and additional experiment results provided by the authors.
>
> I understand that the method does not require identification annotation to be trained. It works by predicting a detection distribution map at the next time step instead of a map of identities associated between two time steps.
>
> The additional experimental evidence added in Appendix Sec A - C are really helpful. Given the same detections (by YOLOX detector), the proposed method shows close performance on MOT17 / MOT20 as ByteTrack. I recognize that the new evidence improves the significance of the proposed method's quantitative performance.
>
> I agree with the intention expressed in `Comment on Benchmark Choice` by the authors in the rebuttal. I didn't ask SOTA performance on every benchmark. The marginal difference after the decimal point is not what I expected to justify whether the proposed method is great or not. I asked for the results on different datasets due to my concern about the generalizability of the proposed method:
>
> To be precise, there are two lines of work:
>
> (1) for appearance-matching-based methods, the similarity of pixels enclosed by bounding boxes is the main clue to associate targets, therefore, a pair of two frames is enough to associate targets.
>
> (2) However, for motion-based tracking methods (this paper and ByteTrack both focus on this genre), the generalizability of the method and the robustness to complex motion patterns is doubtful to me if the method only uses target position information from two consecutive frames.
>
> If we train the model on high-frame-rate videos where the targets mostly move in a linear pattern (as on MOT17 and MOT20) and the input is only target positions on two consecutive frames, what I could expect would be a variant of nearest matching because the model has no idea about the targets' motion intention before timestep t. Kalman filter-based methods solve this problem by maintaining the velocity information in the Kalman filter parameter stack which is derived by accumulating all previous observed motion of the corresponding target.
>
> To answer this question, I asked for the results on two scenarios: (1) highly non-linear motion cases, such as DanceTrack, and (2) low-frame-rate cases, such as KITTI. The new results on MOT20 can partially answer the question and the results in Figure A.5 is also helpful. However, that is not enough in my opinion: The failure of Kalman filter-based methods under a low frame rate is clearly expected because they rely on the linear motion assumption which can hold only under a high enough frame rate by approximating any motion to be linear. So the results in Figure A.5 are not surprising. I did try comparing ByteTrack with the nearest neighbor matching on MOT17 at a low frame rate and ByteTrack also lost the competition.
>
> However, Kalman filter-based methods have proven their effectiveness under complicated motion scenarios, such as on KITTI and DanceTrack when the frame rate is high enough (of course, the relatively low frame rate on KITTI makes some challenges to them). This is my main concern about the generalizability and robustness of the proposed method. Under scenarios where targets have complicated motion and they are crowded (DanceTrack), predicting a pair of detection presence maps seems not enough to me because a naive nearest neighbor matching is not enough anymore. If the proposed method can be demonstrated superior to nearest neighboring matching + IoU matching at all the scenarios, that would be strong evidence to suggest the significance of the proposed method.

---

> > ### Author Response · Authors · 2024-10-21
> > **Further clarification: Visual Features, Non-Linear motion, Motion to Nearest**
> >
> > We sincerely appreciate your thorough feedback and the opportunity to clarify our method's positioning and capabilities.
> >
> > ## Clarification on Method Classification
> >
> > You've categorized the field into appearance-matching and motion-based methods. We'd like to emphasize that our approach bridges these categories by leveraging aspects from both:
> >
> > 1. While we estimate motion between consecutive frames, our model incorporates rich visual features:
> >       - In the multi-view scenario, it processes features from complete image pairs.
> >       - For single-view cases, it uses visual features from detected bounding boxes.
> >
> > 2. Unlike pure appearance-based methods (e.g., Re-ID):
> >       - We don't rely on identity annotations.
> >       - We don't explicitly enforce appearance-based similarity.
> >
> > 3. However, by providing access to appearance-based features, our model can leverage this information for more accurate motion prediction.
> >
> > We've included an ablation study in Section A.4 and Table 1, demonstrating the significant benefit of using appearance features on MOT17. This is evidence of our model's ability to estimate complex motion patterns effectively.
> >
> > ## Addressing Non-linear Motion and Kalman Filter Comparison
> >
> > Regarding your concerns about highly non-linear motion and comparison with Kalman filter:
> >
> > 1. We draw your attention to the experiment in Section A.1, where we present an ablated version of ByteTrack without any motion model. This baseline reveals that, in the ByteTrack's context, the Kalman filter has minimal impact on overall tracking performance.
> >
> > 2. We propose two explanations for these results:
> >       a) ByteTrack's use of IoU matching provides some robustness to misalignment.
> >       b) At low frame rates, as you noted, trajectories become non-linear, which reduces the Kalman filter accuracy.
> >
> > 3. Our motion estimation approach doesn't suffer from these limitations, significantly outperforming ByteTrack at low frame rates while remaining competitive at higher rates.
> >
> > ## Superiority to Nearest Neighbor and IoU Matching
> >
> > We share your concern about demonstrating our method's superiority to nearest neighbor and IoU matching. We've addressed this extensively in our appendix:
> >
> > 1. Section A.2.1: We replace the Kalman filter by nearest neighbor motion estimation. Figure A.2 shows that while this performs better than original ByteTrack at low frame rates, our method significantly outperforms both baselines.
> >
> > 2. Section A.2.2: We conduct similar experiments using Hungarian matching for motion estimation (Figure A.3), reaching the same conclusion.
> >
> > These experiments provide strong evidence that our model is not simply predicting motion to the nearest neighbor, but is learning a more sophisticated motion estimation strategy.
> >
> >
> > ### DanceTrack and KITTI Datasets
> >
> > Regarding your suggestions for additional experiments on DanceTrack and KITTI:
> >
> > 1. For DanceTrack, we anticipate similar results to MOT20 and MOT17. Due to the high frame rate (20 FPS), the motion would likely be locally linear. Moreover, ByteTrack's robustness to small motions due to IoU matching suggests that motion prediction would offer minimal benefits in this scenario.
> >
> > 2. KITTI: As mentioned in our response to Reviewer 81nZ, we have narrowed the scope of our paper to focus solely on people tracking. Since the KITTI tracking benchmark includes cars, we believe it is outside the current scope of our work. Additionally, we have already provided results at low frame rates on three datasets, which we hope sufficiently demonstrate our method's effectiveness in such scenarios.
> >
> > We appreciate your thorough evaluation of our work and have already included results on MOT20 as a show of good faith. We hope that our existing experiments and clarification above have addressed your concerns regarding the value of additional experiments on DanceTrack and KITTI.
> >
> > However, if you still consider these experiments crucial to validating our methods, we are willing to conduct them. Please note that due to the upcoming CVPR deadline (November 15th), we would not be able to complete these experiments before then. We remain open to discussing the best path forward to address any remaining concerns you may have.

---

> > > ### Comment · Reviewer_XLcJ · 2024-11-02
> > >
> > > Thanks for the detailed reply again.
> > >
> > > The authors' feedback has addressed most of my concerns and clarified some details.
> > >
> > > Yes, that would be good and very helpful to provide experiment results on the mentioned additional datasets because they can provide more information about the proposed method, especially on DanceTrack. As mentioned in `Clarification on Method Classification`, the proposed method uses visual features from the detected boxes. In this case, it would be helpful to see whether it can handle the similar appearance challenge on DanceTrack.
> > >
> > > Except for this, I recognize that the authors propose an interesting method for multi-object tracking that needs a week form of supervision information which is beneficial to the community.

---

### Review · Reviewer_81nZ · 2024-09-12

**Summary Of Contributions:**

This manuscript presents a method to train models for multi-object tracking in multi-view data. They build on the tracking by detection paradigm: A detector generates heatmaps for the objects of interest (people in this manuscript) and a post processing helps create tracks out of a sequence of these heatmaps.

Key idea: In addition to training a detector, they also predict motion offsets between time-steps, which are used in the loss function as follows. They use the spatial transformer networks strategy to differentiably warp the detections using the predicted motion offsets. These warped detection heatmaps are evaluated against the ground truth heatmaps from the future time-step.

The main claims are:
1. Doing so circumvents the need for identity or other non-detection annotations which are typically very expensive to obtain.
2. Doing so also makes the model more robust to frame rate reduction (in other words, the model can handle larger displacements between time-steps).

**Audience:**

Yes

**Broader Impact Concerns:**

People tracking is a very sensitive topic prone to misuse by state and non-state powers. The authors themselves acknowledge the sensitive application of surveillance. A broader impact section is needed for this paper.

Additionally, in my humble opinion, it is high time the literature in this field started documenting model fairness. For example, underlying detection models are not always equally accurate across race, skin colour, sex, hairstyle, glasses etc. How does that effect the tracking performance? Is tracking performance worse or better for certain groups or intersections thereof?

This paper studies model robustness with changing frame rate. This is a good step forward. But robustness of these models in other dimensions would be helpful in addressing broader impact concerns. For example effect of weather, indoors vs outdoors, amount of sunlight etc.

**Claims And Evidence:**

No

**Requested Changes:**

Already documented in the previous section.

**Strengths And Weaknesses:**

### Strengths

- The method is simple and easy to understand. Although the memory usage is quadratic with number of heatmap pixels forcing the method to be used in a sliding window manner.
- The Table 3 "motion offset evaluation" experiment is particularly nice in the light of TMLRs focus on solid experimentation backing the claims made in the paper. In this experiment they try to use motion offsets directly predicted by RAFT (optical flow model) instead of teaching the model to predict those. They also use ground truth motion offsets to have a supervised baseline. Their method is much better than the RAFT based ablation and comes close to the supervised baseline.

### Weaknesses
- The manuscript is scoped to cover 'object' motion but all the experiments are scoped to cover 'people' motion which is a subset. Even though the method is not specialized for people, because of the limited experiments, the title and the text need to be adjusted to reflect this more limited scope.
- The experiments demonstrate that the proposed motion offset based method performs competitively against methods that use more supervision eg. identity supervision. But this is not the same as showing that the need for identity supervision is circumvented, which is the idea I got after reading the introduction. In order to demonstrate that the need for identity supervision is circumvented, one needs to run the ablation "Ours + Identity supervision" and show that this is roughly on par with "Ours". Alternatively it must be made clearer that this method does not intend to replace identity supervision, which may as well be orthogonal to the contributions of this paper.
- Are the FPS vs metric value plots in figure 1 and figure 5 aligned? It will help the reader if these two plots are made equivalent in terms of metrics, loss, datasets, models etc. That way the reader can understand, as to what extent the problem initially motivating the manuscript has been solved.
- There are no error bars reported for any of the results. In case this is computationally too expensive, please at least provide error bars for the numbers for "Ours" in Table 1.
- Please clarity where the ground truth offsets for section 4.2.2 are obtained from. Are these part of the dataset?
- The proposed method includes several loss terms and many hyper-parameters. A table summarizing all the hyper-parameters will greatly assist readers trying to benefit from this manuscript. I cannot confirm whether all the hyper-parameters have been documented in the manuscript without such a table.

---

> ### Author Response · Authors · 2024-09-24
> **Clarifying Identity Supervision and Motion Annotations: Response to Review**
>
> ### Scope
> - The title of the paper has been changed to "No Identity, no problem: Motion through detection for people tracking" to better reflect the scope of this work.
>
> ### Identity Supervision
> - There seems to be a misunderstanding regarding our use of the term "identity annotations." By this, we mean instances where someone marks detections in consecutive frames as belonging to the same person. Throughout the paper, "identity annotations" was used interchangeably with "motion annotations," as it is always possible to convert ground-truth motion offsets to identity labels and vice versa. Our method does away with the need for this.
>
> - However, you are correct when pointing out that identity supervision is broader than motion supervision, encompassing the large field of re-identification. We did not anticipate this misinterpretation, and it was not our intent to compare our method to such approaches, which are indeed orthogonal to our contribution. We have clarified this in the manuscript.
>
> ### Introduction Figure
> - We have updated Figure 1 to better align its x-axis with the subsequent figures.
>
> ### Error Bars
> - As it is too costly to compute error bars for all experiments, we have included the standard deviation for our results on the WILDTRACK dataset in Table 1.
>
> ### Ground-truth Motion Offsets
> - The ground-truth offsets used to train the fully supervised variant of our model are derived from the identity labels present in the WILDTRACK dataset.
>
> ### Hyperparameters
> - A table summarizing the hyperparameters used by our method has been added to the appendix, section D.
>
> ### Broader Impact
> - A broader impact section has been included in the manuscript, section 5.

---

> > ### Comment · Reviewer_81nZ · 2024-10-23
> >
> > First of all, I apologise for the delay. I fell ill and couldn't get back to you earlier, also because of the ICLR deadline.
> >
> > Fig. 1 vs Fig. 5: muSSP in Fig. 1 at frame rate 0.5 achieves MoTA of 0.8 which is far above the MoTA for ByteTrack and Ours reported in Fig 5. Are these different datasets?
> >
> > Thank you for addressing all my concerns.

---

> > > ### Author Response · Authors · 2024-10-23
> > > **Clarification on Figure 1**
> > >
> > > Figure 1 was computed using the WILDTRACK dataset and is meant to illustrate the challenges faced by the association algorithm at low framerate. To separate this from potential detection failures, we used ground truth bounding boxes in this figure. As a result, the performance reported here (e.g., muSSP achieving a MoTA of 0.8 at 0.5 fps) is not directly comparable to later figures or tables. We will update the caption of Figure 1 to make this clearer.
> > >
> > > Thank you for your feedback. We are available should you have any further questions.

---

### Author Response · Authors · 2024-09-24
**General comment**

We would like to sincerely thank the reviewers for their valuable feedback and insightful comments. We have carefully considered each point and have revised the paper accordingly, including the suggested improvements and additional analyses. The revisions are highlighted in red for your convenience. We appreciate the time and effort invested in reviewing our work and look forward to addressing any further concerns.

---

### Decision · Action_Editor_2NXj · 2024-11-11

**Recommendation:** Accept as is

**Comment:**

All three reviewers recommend that the paper should be accepted and are happy with the revisions. There are, to the best of my knowledge, no pending concerns that would warrant a revision since the authors have addressed all of the reviewer concerns in their response to the reviews. I therefore recommend accepting the paper as-is (i.e., with the authors simply updating the paper so that the changes are no longer in red).

**Audience:**

Yes. The topic is of clear interest to a substantial number of people in the TMLR community.

**Claims And Evidence:**

After revision, all reviewers agree that the paper's claims are well-supported.